# SelecMix: Debiased Learning by Contradicting-pair Sampling

**Inwoo Hwang**[1]    **Sangjun Lee**[1]    **Yunhyeok Kwak**[1]    **Seong Joon Oh**[3]
**Damien Teney**[4]    **Jin-Hwa Kim**[*12]    **Byoung-Tak Zhang**[*1]
[1]AI Institute, Seoul National University    [2]NAVER AI Lab
[3]University of Tübingen    [4]Idiap Research Institute

## Abstract

Neural networks trained with ERM (empirical risk minimization) sometimes learn unintended decision rules, in particular when their training data is biased, i.e., when training labels are strongly correlated with undesirable features. To prevent a network from learning such features, recent methods augment training data such that examples displaying spurious correlations (i.e., *bias-aligned* examples) become a minority, whereas the other, *bias-conflicting* examples become prevalent. However, these approaches are sometimes difficult to train and scale to real-world data because they rely on generative models or disentangled representations. We propose an alternative based on mixup, a popular augmentation that creates convex combinations of training examples. Our method, coined SelecMix, applies mixup to *contradicting pairs* of examples, defined as showing either (i) the same label but dissimilar biased features, or (ii) different labels but similar biased features. Identifying such pairs requires comparing examples with respect to unknown biased features. For this, we utilize an auxiliary contrastive model with the popular heuristic that biased features are learned preferentially during training. Experiments on standard benchmarks demonstrate the effectiveness of the method, in particular when label noise complicates the identification of bias-conflicting examples.

## 1   Introduction

The inductive biases contributing to the success of deep neural networks (DNNs) can sometimes limit their capabilities for out-of-distribution (OOD) generalization. DNNs are prone to learn simple, linear predictive patterns from their training data, sometimes ignoring more complex but important ones [30, 32]. It has been suggested that the simplest correlations in the data are often spurious [7]. A DNN relying on such simple, spurious patterns will therefore display poor OOD generalization. Spurious correlations in a dataset are often the result of a selection bias, and such datasets are therefore said to be *biased*. This paper is about debiased learning, also known as debiasing, i.e., methods that prevent a network from relying on spurious correlations when trained on a biased dataset.

Biased datasets typically contain a majority of so-called *bias-aligned* examples and a minority of *bias-conflicting* ones. In bias-aligned examples, ground truth labels are correlated with both robust and biased features.[2] In bias-conflicting examples, labels are correlated only with robust features. Clearly, the issues of models trained on biased datasets stem from the prevalence of bias-aligned samples. Various approaches for debiased learning encourage models to ignore biased features. Since

---

[*]Corresponding authors.

[2]A feature is biased if it displays a pattern that is statistically predictive of the labels over the dataset, though not necessarily on every example. For instance, a blue background may be present in most (but not all) images of birds. These images are said to be *bias-aligned*.

36th Conference on Neural Information Processing Systems (NeurIPS 2022).

the identification of biased features from i.i.d. data is ill-defined, it requires additional assumptions, or supervision from heterogeneous (non-i.i.d.) training samples [28].

In this work, we approach debiased learning with the assumption that biased features are "easier to learn" than robust ones, meaning that they are incorporated in the model earlier during training [29, 30] [3]. Existing works based on this heuristic typically train two models: (i) an auxiliary model that purposefully relies on biased features, and (ii) the desired debiased model. The auxiliary one guides the training of the debiased one [24, 27]. For example, Nam et al. [24] trains the auxiliary model with a *generalized cross-entropy* (GCE) loss [40] that strengthens its reliance on biased, easy-to-learn features. The training of the debiased model either augments the data with novel bias-conflicting examples [15, 20] or upweights existing ones [22, 24]. On the one hand, upweighting-based methods are simple but their debiasing capabilities are limited when bias-conflicting examples are scarce. On the other hand, augmentation-based methods rely on carefully tuned generative models or disentangled representations that are difficult to apply to real-world data.

We propose a simple and effective method based on mixup [37], a popular data augmentation method that creates convex combinations of randomly-chosen pairs of examples and their labels. Our method, coined SelecMix, is an application of mixup to selected *contradicting pairs* of examples to generate new bias-conflicting examples. We define contradicting pairs as having either (i) the same ground truth label but dissimilar biased features, or (ii) different labels but similar biased features. To compare examples with respect to their biased features and thereby identify the contradicting pairs, we train an auxiliary contrastive model with a novel *generalized supervised contrastive* (GSC) loss that amplifies the reliance on easy-to-learn features. As a result, feature clustering in the embedding space serves as a good indicator of the similarity of the biased features. We train the auxiliary model and the desired debiased model simultaneously. The auxiliary model identifies contradicting pairs, while the debiased model is trained on data augmented by SelecMix. Compared to past approaches, our method generates bias-conflicting examples without generative models or disentanglement, while implicitly upweighting existing ones since they are frequently selected for the mixup.

We evaluate our method on standard benchmarks for debiasing. Experimental results suggest that SelecMix consistently outperforms prior methods, especially when bias-conflicting samples are scarce. In addition, our method maintains its superior performance in the presence of label noise that complicates the identification of bias-conflicting examples. Ablation studies show advantages of both (i) the selective mixup strategy compared to other mixup variants, and (ii) the GSC loss, which strengthens the reliance on biased features and allows measuring the similarity of examples with respect to the biased features.

## 2  Related work

**Debiasing with known forms of bias or bias labels.**  Early works on debiasing assume some knowledge about the bias. Some methods require that each training example is provided with a *bias label*, i.e., a precise value for its biased features [11, 14, 21, 26, 31]. Other methods use knowledge of the general form of the bias, such as color or texture in images. This information is typically used to design custom architectures [1, 5, 34]. For example, ReBias [1] uses a BagNet [4] as an auxiliary model because it focuses on texture, which is assumed to be the biased feature. The auxiliary model guides the training of a debiased model robust to unusual variations in texture.

**Debiasing with the easy-to-learn heuristic.**  A number of recent works assume that biased features are learned more quickly than robust ones [20, 24]. A popular approach is to train an auxiliary model that intentionally relies primarily on the easy-to-learn biased features, e.g., with the GCE loss [40]. The auxiliary model guides the training of a debiased model that focuses on other, presumably non-biased features. For example, LfF [24] learns biased and debiased models simultaneously. Bias-conflicting training examples are identified based on the relative losses from the biased and debiased models, then upweighted for the training the debiased model. BiaSwap [15] trains an image translation model to generate new bias-conflicting training examples. Building on LfF, DFA [20] disentangles robust and biased features and swaps them randomly for augmentation. Disentanglement is however an ill-posed problem in itself [23] and a challenge with real-world data. In contrast, our

---

[3]See [38] for a discussion of the disputed relevance of this assumption to real-world data.

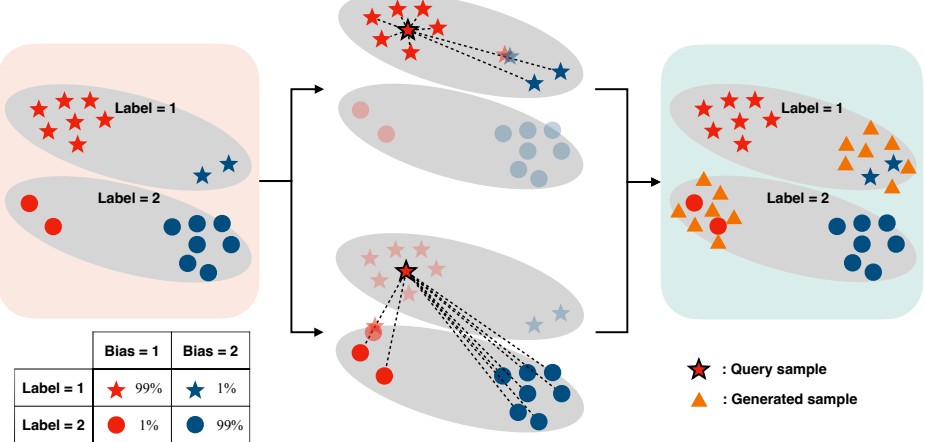

Figure 1: Overview of the proposed SelecMix. A gray ellipse with the symbols represents a sharing label on the embedding space. The selective mixup is applied to the pairs of samples (dashed lines) having **(top)** the same label but dissimilar biased features, and **(below)** the different labels but similar biased features (Sec. 3.3). The orange triangles of the top and below ellipses represent the generated samples by the previous two procedures, respectively. The legend illustrates the distribution of the exemplary dataset of two labels and two bias labels.

method augments bias-conflicting data by mixing existing examples, without generative models nor disentangled representations.

# 3 Method

We first describe the use of mixup as a debiasing strategy, while assuming that each training example is (unrealistically) provided with bias labels (Sec. 3.1). Then, we move to a realistic scenario where bias labels are not available. We introduce an auxiliary contrastive model that identifies the biased features, which are assumed to be "easier to learn" than robust ones. This allows comparing examples with respect to these biased features (Sec. 3.2). Finally, we describe the complete SelecMix method that combines the selective mixup with our auxiliary biased model (Sec. 3.3).

## 3.1 Mixup for augmenting bias-conflicting samples

In a highly biased dataset, bias-conflicting samples constitute only a small fraction of the training data. This is the root issue in the cases we consider. The goal is thus to increase the fraction of bias-conflicting examples in the training data, which will then reduce the reliance of the learned model on biased features. Our idea is to use mixup [37] to augment the existing pool of bias-conflicting examples. Mixup is a popular augmentation method [2, 16, 36] known to improve various measures of robustness [25, 39]. It constructs convex combinations of pairs of examples and their labels:

$$(\widetilde{x}_i, \widetilde{y}_i) \quad \leftarrow \quad \big(\lambda\, x_i + (1-\lambda)\, x_j, \ \lambda\, y_i + (1-\lambda)\, y_j\big), \tag{1}$$

where $(x_i, y_i)$ and $(x_j, y_j)$ are two original training examples (e.g., image and one-hot label vector) and $\lambda$ is a random mixing coefficient, $\lambda \sim U[0, 1]$.

Assuming for now that bias labels are available, we could generate bias-conflicting examples by applying mixup on pairs having either (i) the same ground truth label but different bias labels or (ii) different labels but the same bias label. Any such pair includes at least one bias-conflicting example, so that mixup generates additional ones as long as this original example is assigned a higher mixing weight in Eq. (1). An overview of this mixup strategy is shown in Fig. 1. Next, we describe how to identify such desired pairs when bias labels are not available.

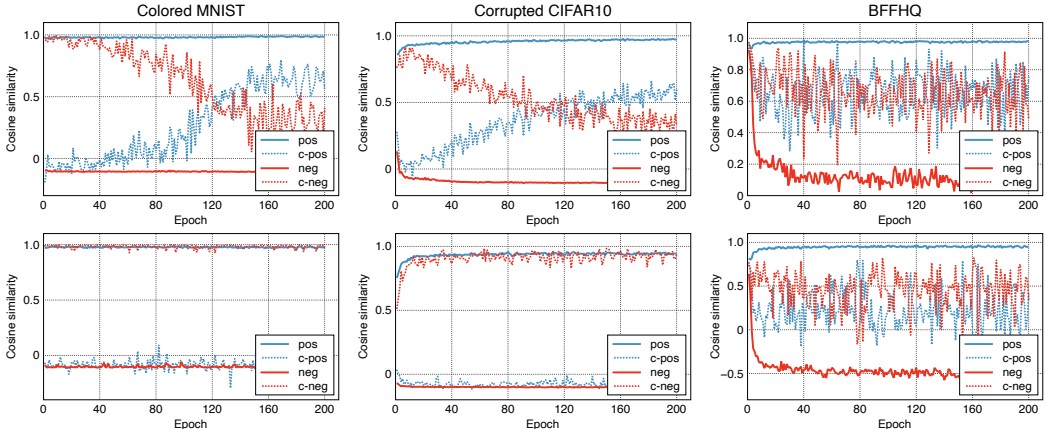

Figure 2: Illustration of the similarity of positive, negative, and contradicting pairs when trained with **(top)** the SC and **(bottom)** the proposed GSC losses. The solid line shows the average cosine similarity of (i) the pairs with the same label (*positives*) and (ii) the pairs with different labels (*negatives*). The dotted line represents (iii) the pairs with the same label but different bias labels (*contradicting positives*) and (iv) the pairs with the different labels but the same bias label (*contradicting negatives*). **Observations:** As training proceeds with the SC loss, the similarity of contradicting positives increases while it decreases for the contradicting negatives. In contrast, the proposed GSC loss amplifies the reliance on the biased features, thus the clustering in embedding space remains a good indicator of the similarity of the biased features during the whole training process.

## 3.2 Replacing bias labels with an auxiliary model

We utilize an auxiliary model to compare training examples with respect to their biased features. We assume that biased features are easier to learn than robust ones, because they are involved in simpler (e.g., linear) predictive patterns. Our auxiliary model is trained to rely primarily on biased features. We train the auxiliary model with a contrastive objective [6, 10, 13]. This is known to induce a clustering in embedding space (better than standard cross-entropy) that reflects the similarity of training examples in terms of the learned features. These are *biased* features by our assumption, such that the clustering reflects the similarity of examples w.r.t. their (unknown) bias labels. We use the supervised contrastive (SC) loss of Khosla et al. [13]:

$$\mathcal{L}_{SC} = -\sum_{i \in \mathcal{B}} \frac{1}{|\mathcal{P}_i|} \sum_{k \in \mathcal{P}_i} \log p_{i,k}, \quad \text{where} \quad p_{i,k} = \frac{\exp\left(\boldsymbol{z}_i^\mathsf{T} \boldsymbol{z}_k / \tau\right)}{\sum_{j \in \mathcal{B} \setminus \{i\}} \exp\left(\boldsymbol{z}_i^\mathsf{T} \boldsymbol{z}_j / \tau\right)}, \tag{2}$$

where $\boldsymbol{z}_i$ is the normalized embedding of image $x_i$, $\mathcal{B} = \{1, 2, ..., B\}$ is the set of indices in the current mini-batch, $\mathcal{P}_i = \{k \in \mathcal{B} \setminus \{i\} \mid y_i = y_k\}$ is the set of positive examples relative to the example $i$ (i.e., with the same label), and the scalar $\tau$ is a temperature hyperparameter.

As an experiment to confirm that the clustering of training examples in embedding space is indeed based on biased features, we train the auxiliary model on the Colored MNIST, the Corrupted CIFAR-10, and the BFFHQ datasets (see Appendix A.1) with the SC loss and compute the cosine similarity $\boldsymbol{z}_i^\mathsf{T} \boldsymbol{z}_j$ of the embeddings of all pairs of examples. Fig. 2 shows that the examples are clustered according to the biased features early in the training. In other words, predictive patterns involving biased features are learned faster than those involving robust features, as desired. Since the higher cosine similarity $\boldsymbol{z}_i^\mathsf{T} \boldsymbol{z}_j$ implies a high probability $p_{i,j}$, we interpret it as the *likelihood* of the pair $(i, j)$ having similar biased features.

To further amplify the reliance of the auxiliary model on the biased features, we define the *generalized SC* (GSC) loss as follows:

$$\mathcal{L}_{GSC} = -\sum_{i \in \mathcal{B}} \frac{1}{|\mathcal{P}_i|} \sum_{k \in \mathcal{P}_i} \hat{p}_{i,k}^q \log p_{i,k}, \tag{3}$$

**Algorithm 1** SELECMIX

1: **Input**: batch $\mathcal{B} = \{(x_i, y_i)\}_{i=1}^B$, biased model $g_\phi$
2: **Output**: batch $\widetilde{\mathcal{B}} = \{(\tilde{x}_i, \tilde{y}_i)\}_{i=1}^B$
3: Sample $p \sim \mathrm{U}[0,1]$, $\widetilde{\mathcal{B}} = \varnothing$
4: **for** $i = 1, \cdots, B$ **do**
5:    Sample $\lambda \sim \mathrm{U}[0,1]$ and $\lambda \leftarrow \min(\lambda, 1-\lambda)$
6:    if $p > 0.5$: $k = \underset{j \in \mathcal{P}_i}{\operatorname{argmin}} \, g_\phi(x_i)^\mathsf{T} g_\phi(x_j)$
7:    else:    $k = \underset{j \in \mathcal{N}_i}{\operatorname{argmax}} \, g_\phi(x_i)^\mathsf{T} g_\phi(x_j)$
8:    $(\tilde{x}_i, \tilde{y}_i) \leftarrow (\lambda x_i + (1-\lambda)x_k, y_k)$
9:    $\widetilde{\mathcal{B}} \leftarrow \widetilde{\mathcal{B}} \cup \{(\tilde{x}_i, \tilde{y}_i)\}$
10: **end for**

**Algorithm 2** Training with SELECMIX

1: **Input**: a dataset $\mathcal{D} = \{(x_i, y_i)\}$, a model $f_\theta$, a biased model $g_\phi$, the number of iterations $T$
2: **Output**: a debiased model $f_\theta$
3: Initialize $\theta$ and $\phi$
4: **for** $t = 1, \cdots, T$ **do**
5:    Draw a batch $\mathcal{B} = \{(x_i, y_i)\}_{i=1}^B$ from $\mathcal{D}$
6:    $\widetilde{\mathcal{B}} = \text{SELECMIX}(\mathcal{B}, g_\phi)$
7:    Update $\theta$ with $\mathcal{L}_{\text{CE}}(\widetilde{\mathcal{B}})$
8:    Update $\phi$ with $\mathcal{L}_{GSC}(\mathcal{B})$      {Eq. (3)}
9: **end for**

where $\hat{p}_{i,k}^q$ is a scalar having the same value as $p_{i,k}^q$, meaning that the gradient is not back-propagated through it. The term $\hat{p}_{i,k}^q$ assigns a higher weight to sample pairs with a high probability $p_{i,k}$ and thus amplifies the reliance on the biased features. We discuss the relationship between the GCE and GSC losses in Appendix C.2.

### 3.3 Complete proposed method: selective mixup with biased embedding space

We now have an auxiliary model for quantifying the similarity of the training examples in terms of the biased features. We use it to apply mixup on *contradicting pairs*, which have either (i) the same label but dissimilar biased features (*contradicting positives*) or (ii) different labels but similar biased features (*contradicting negatives*).

**Contradicting positives.** For each instance $(x_i, y_i)$ in the current mini-batch (i.e., the "query"), we pick another one $(x_k, y_k)$ with the lowest similarity (measured in the space of their embeddings produced by the auxiliary model) among the set of positive examples (i.e., with the same label as $x_i$):

$$k = \underset{j \in \mathcal{P}_i}{\operatorname{argmin}} \, p_{i,j} = \underset{j \in \mathcal{P}_i}{\operatorname{argmin}} \, \boldsymbol{z}_i^\mathsf{T} \boldsymbol{z}_j, \quad \text{where } \mathcal{P}_i = \{j \in \mathcal{B} \setminus \{i\} \mid y_i = y_j\}, \tag{4}$$

where $\mathcal{B} = \{1, 2, ..., B\}$ is the set of the sample indices in the current mini-batch, and $\boldsymbol{z}_i$ and $\boldsymbol{z}_j$ are the normalized embeddings produced by our auxiliary model $g_\phi$, i.e., $\boldsymbol{z}_i = g_\phi(x_i)$. Since we select the pair among the set of positives, the training CE loss of the mixed example is $l(\tilde{x}_i, \tilde{y}_i) = l(\lambda x_i + (1-\lambda)x_k, \lambda y_i + (1-\lambda)y_k) = l(\lambda x_i + (1-\lambda)x_k, y_k)$. Considering that most examples are bias-aligned in the training set, the query $(x_i, y_i)$ is also likely to be bias-aligned. In addition, since the query $x_i$ and the selected example $x_k$ have the same label but dissimilar biased features, it is also likely that the biased features of $x_k$ are not correlated with the label, i.e., the selected example $(x_i, y_i)$ is likely to be bias-conflicting. Thus, to effectively generate an example that contradicts the prediction based on biased features, we sample $\lambda \sim U[0,1]$ and assign the smaller value among $\lambda$ and $1-\lambda$ to $x_i$ and the larger one to $x_k$.

**Contradicting negatives.** For each query $(x_i, y_i)$, we pick another one with the highest similarity among the set of negative examples:

$$k = \underset{j \in \mathcal{N}_i}{\operatorname{argmax}} \, p_{i,j} = \underset{j \in \mathcal{N}_i}{\operatorname{argmax}} \, \boldsymbol{z}_i^\mathsf{T} \boldsymbol{z}_j, \quad \text{where } \mathcal{N}_i = \{j \in \mathcal{B} \mid y_i \neq y_j\}. \tag{5}$$

Similarly, we sample $\lambda \sim U[0,1]$ and let $\lambda \leftarrow \min(\lambda, 1-\lambda)$. In standard mixup, the training loss of the mixed sample is: $l(\tilde{x}_i, \tilde{y}_i) = l(\lambda x_i + (1-\lambda)x_k, \lambda y_i + (1-\lambda)y_k) = \lambda \cdot l(\lambda x_i + (1-\lambda)x_k, y_i) + (1-\lambda) \cdot l(\lambda x_i + (1-\lambda)x_k, y_k)$. However, considering that (i) the query $(x_i, y_i)$ is likely to be bias-aligned, and (ii) the pair $(x_i, x_k)$ shares similar biased features, the first term $\lambda \cdot l(\lambda x_i + (1-\lambda)x_k, y_i)$ acts as bias-aligned example since the biased feature shared with $x_i$ and $x_k$ is predictive of the label $y_i$. Thus, rather than interpolating the label, we simply assign $\tilde{y}_i \leftarrow y_k$. The pseudo-code of the proposed algorithm is presented in Alg. 1 and Alg. 2.

**Intuitive interpretation.** Unlike standard mixup, we assign the label of the generated sample to the label of the selected sample, which is likely to be bias-conflicting. Therefore, the proposed method

can be viewed as **generating new bias-conflicting samples by injecting bias-aligned samples as noise to existing bias-conflicting samples**. The method can also be interpreted as implicitly upweighting existing bias-conflicting samples, since they are frequently chosen for the mixup pair.

# 4 Experiments

We now validate the effectiveness of the proposed method on debiasing benchmarks. We also evaluate it under the presence of label noise, which is a challenging but realistic scenario, yet less explored so far (Sec. 4.1). We also provide a detailed analysis of each component of our method: (i) the auxiliary contrastive model (Sec. 4.2) and (ii) the selective mixup strategy (Sec. 4.3).

**Datasets.** The Colored MNIST is a modified MNIST [19] that consists of colored images of ten digits where each digit is correlated with the color (e.g., most of the images of "0" are colored with red). Here, the label is a digit (i.e., $0 \sim 9$) and the biased feature is color. The Corrupted CIFAR10 is constructed by applying different types of corruptions to the corresponding objects in the original CIFAR-10 [18] dataset (e.g., most of the images of dogs are corrupted with GAUSSIAN BLUR noise). The Biased FFHQ (BFFHQ) [20] is constructed based on the real-world dataset FFHQ [12], where the label is age and the biased feature is gender. The ratio of bias-conflicting samples in the training set is $\alpha \in \{0.5\%, 1\%, 2\%, 5\%\}$ for {Colored MNIST, Corrupted CIFAR10} and $\alpha = 0.5\%$ for BFFHQ. All datasets are available in the official repository of DFA [20]. We defer the detailed description of the datasets in Appendix A.1.

**Baselines.** To evaluate the effectiveness of our method in debiasing, we compare it with the prior methods LfF [24] and DFA [20] which also rely on the easy-to-learn property of the biased features. LfF trains an auxiliary model with the GCE loss to amplify its reliance on the biased features, then reweights examples for training a debiased model. DFA disentangles the biased and robust features with a similar principle as LfF, then augments the data for training a debiased model by swapping the biased features across examples. We also include EnD [31], ReBias [1], and HEX [34]. EnD leverages explicit bias labels. ReBias and Hex are designed for a specific, known form of biased features such as color and texture in images.

Table 1: Main results. (*) Denotes methods tailored to predefined forms of bias, (°) methods using bias labels, and ($\dagger$) methods relying on the easy-to-learn heuristic. Numbers for HEX are from [20].

| Dataset | Ratio (%) | Vanilla | HEX [*] [34] | ReBias [*] [1] | EnD [°] [31] | LfF [$\dagger$] [24] | DFA [$\dagger$] [20] | V+Ours [$\dagger$] | L+Ours [$\dagger$] |
|---|---|---|---|---|---|---|---|---|---|
| Colored MNIST | 0.5 | $35.71_{\pm0.83}$ | $30.33_{\pm0.76}$ | $\mathbf{71.42}_{\pm1.41}$ | $56.98_{\pm4.85}$ | $63.86_{\pm2.81}$ | $67.37_{\pm1.61}$ | $\underline{70.47}_{\pm1.66}$ | $70.00_{\pm0.52}$ |
| | 1.0 | $50.51_{\pm2.17}$ | $43.73_{\pm5.50}$ | $\mathbf{86.50}_{\pm0.97}$ | $73.83_{\pm2.09}$ | $78.64_{\pm1.51}$ | $80.20_{\pm1.86}$ | $\underline{83.55}_{\pm0.42}$ | $82.80_{\pm0.71}$ |
| | 2.0 | $65.40_{\pm1.63}$ | $56.85_{\pm2.58}$ | $\mathbf{92.95}_{\pm0.21}$ | $82.28_{\pm1.08}$ | $84.95_{\pm1.71}$ | $85.61_{\pm0.76}$ | $87.03_{\pm0.58}$ | $\underline{87.16}_{\pm0.62}$ |
| | 5.0 | $82.12_{\pm1.52}$ | $74.62_{\pm3.20}$ | $\mathbf{96.92}_{\pm0.09}$ | $89.26_{\pm0.27}$ | $89.42_{\pm0.65}$ | $89.86_{\pm0.80}$ | $91.56_{\pm0.17}$ | $\underline{91.57}_{\pm0.20}$ |
| Corrupted CIFAR-10 | 0.5 | $23.26_{\pm0.29}$ | $13.87_{\pm0.06}$ | $22.13_{\pm0.23}$ | $22.54_{\pm0.65}$ | $29.36_{\pm0.18}$ | $30.04_{\pm0.66}$ | $\underline{38.14}_{\pm0.15}$ | $\mathbf{39.44}_{\pm0.22}$ |
| | 1.0 | $26.10_{\pm0.72}$ | $14.81_{\pm0.42}$ | $26.05_{\pm0.10}$ | $26.20_{\pm0.39}$ | $33.50_{\pm0.52}$ | $33.80_{\pm1.83}$ | $\underline{41.87}_{\pm0.14}$ | $\mathbf{43.68}_{\pm0.51}$ |
| | 2.0 | $31.04_{\pm0.44}$ | $15.20_{\pm0.54}$ | $32.00_{\pm0.81}$ | $32.99_{\pm0.33}$ | $40.65_{\pm1.23}$ | $42.10_{\pm1.04}$ | $\underline{47.70}_{\pm1.35}$ | $\mathbf{49.70}_{\pm0.54}$ |
| | 5.0 | $41.98_{\pm0.12}$ | $16.04_{\pm0.63}$ | $44.00_{\pm0.66}$ | $44.90_{\pm0.37}$ | $50.95_{\pm0.40}$ | $49.23_{\pm0.63}$ | $\underline{54.00}_{\pm0.38}$ | $\mathbf{57.03}_{\pm0.48}$ |
| BFFHQ | 0.5 | $56.20_{\pm0.35}$ | $52.83_{\pm0.90}$ | $56.80_{\pm1.56}$ | $56.53_{\pm0.61}$ | $65.60_{\pm1.40}$ | $61.60_{\pm1.97}$ | $\mathbf{71.60}_{\pm1.91}$ | $\underline{70.80}_{\pm2.95}$ |

## 4.1 Main results

We apply our method to a vanilla ResNet18 (V+Ours) and to LfF (L+Ours). We use the Colored MNIST, Corrupted CIFAR10, and BFFHQ datasets. As shown in Table 1, our method consistently outperforms baselines, except ReBias [1] on the Colored MNIST. Their bias-capturing model Bag-Net [4] is specifically tailored to color and texture as biased features by relying on local image patches as input. Both DFA and our method augment the pool of bias-conflicting training examples, but DFA's reliance on disentangled representations seems problematic on the more complex datasets. In contrast, our method performs well on all datasets, owing to the simplicity of the mixup strategy. In particular, our method outperforms DFA by a large margin on BFFHQ while the gap is smaller on Colored MNIST where disentanglement is easier. See Appendix A.2 for experimental details.

**Debiasing under the presence of label noise.** Label noise can negatively affect debiasing methods that need to identify bias-conflicting examples. This is because training examples with noisy (i.e., incorrect) labels are difficult to distinguish from the bias-conflicting examples that we wish to identify. Fig. 3 shows that our method maintains better performance under the presence of label noise than

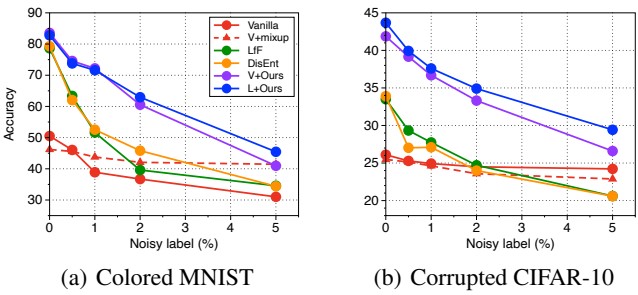

|              (a) Colored MNIST                |              (b) Corrupted CIFAR-10                |

Figure 3: Unbiased accuracy under the presence of label noise.

Table 2: Ablation study of various metrics on the biased embedding space. GT indicates that we used ground truth bias label for SelecMix. The cos and $l_2$ denote cosine distance and $l_2$ distance on the embedding space of the biased model trained with GCE loss, respectively. Ours denote the cosine distance on the embedding space of the proposed biased contrastive model.

| SelecMix | GT | cos | $l_2$ | KL-divergence | Ours |
|---|---|---|---|---|---|
| Contradicting positives | $41.71_{\pm0.85}$ | $40.31_{\pm0.63}$ | $38.56_{\pm1.25}$ | $40.82_{\pm0.94}$ | $\mathbf{42.94}_{\pm0.44}$ |
| Contradicting negatives | $41.70_{\pm0.74}$ | $33.69_{\pm0.97}$ | $34.15_{\pm0.50}$ | $30.99_{\pm0.93}$ | $\mathbf{41.82}_{\pm0.98}$ |
| Both | $43.35_{\pm0.67}$ | $38.05_{\pm0.83}$ | $38.90_{\pm0.96}$ | $39.94_{\pm1.48}$ | $\mathbf{43.68}_{\pm0.51}$ |

baselines. We hypothesize that the robustness of our method comes from the nature of mixup, which is known to improve robustness [39].

## 4.2 Detailed analysis of the biased contrastive model

**Comparison of the biased embedding spaces for measuring bias similarity.** Our auxiliary contrastive model trained with the GSC loss learns the embedding space which reflects the similarity of the examples w.r.t. their biased features. It is then used for SelecMix to identify the contradicting pairs by measuring the cosine similarity. We replace our auxiliary model with the biased model of LfF [24] which is trained with the GCE loss, and use it for SelecMix. For the similarity measures, we used cosine distance and $l_2$ distance in the embedding space learned by their biased model, and the KL-divergence of the softmax outputs of the classification head. As shown in Table 2, our auxiliary contrastive model achieves the best performance. Especially, the performance gap is significant for the contradicting negatives. This supports our claim in Sec. 3.2 that the proposed GSC loss induces the feature clustering in the embedding space w.r.t. the biased features. In contrast, the GCE loss learns the decision boundary and does not explicitly cluster the features, thus selecting the contradicting negatives, i.e., the pairs with different labels but similar biased features, seems to underperform ours.

Table 3: Bias label prediction accuracy.

| Biased model | Corrupted CIFAR-10 | | | | BFFHQ |
| | 0.5% | 1.0% | 2.0% | 5.0% | 0.5% |
|---|---|---|---|---|---|
| LfF [24] | $77.44_{\pm0.94}$ | $73.01_{\pm1.70}$ | $67.00_{\pm0.87}$ | $55.58_{\pm0.06}$ | $51.07_{\pm3.06}$ |
| DFA [20] | $79.03_{\pm1.15}$ | $72.30_{\pm0.71}$ | $64.71_{\pm0.24}$ | $52.01_{\pm0.70}$ | $46.20_{\pm1.00}$ |
| Ours | $\mathbf{95.45}_{\pm0.05}$ | $\mathbf{93.39}_{\pm0.02}$ | $\mathbf{92.89}_{\pm0.06}$ | $\mathbf{87.28}_{\pm0.20}$ | $\mathbf{59.13}_{\pm0.23}$ |

**Bias label prediction of the biased model.** Table 3 shows the accuracy of the bias label prediction of the auxiliary biased model for each method. Our biased contrastive model does not have a classification head since it is trained with a contrastive loss. Thus, we attach a linear classifier on top of the model and fine-tune it. Note that the bias labels are used only for the evaluation. As shown in Table 3, the biased contrastive model shows the best performance. As the ratio of the bias-conflicting samples increases, they degrade the performance. Notice our biased contrastive model is robust to this effect compared with the prior methods relying on the GCE loss.

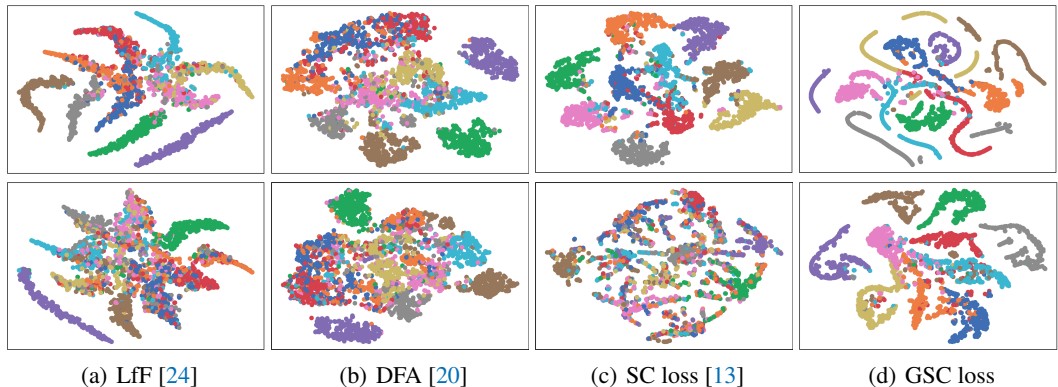

| (a) LfF [24] | (b) DFA [20] | (c) SC loss [13] | (d) GSC loss |

Figure 4: Visualization with t-SNE of features extracted from (a) the biased model of LfF [24], (b) the biased model of DFA [20], (c) the model trained with the SC loss, and (d) our auxiliary contrastive model trained with the GSC loss. **(Top)** $\alpha$=1%. **(Bottom)** $\alpha$=5%.

**Visualization.** Fig. 4 illustrates t-SNE [33] plot of the features extracted from the corresponding models trained on the Corrupted CIFAR10. As shown in Fig. 4(d), which corresponds to the proposed biased contrastive model, we observe that the samples are well clustered on the embedding space of the biased contrastive model. Similar to the results in Table 3, the GCE-based biased models, i.e., LfF and DFA, suffer from the increasing number of bias-conflicting samples.

**Pretraining vs. simultaneous training of the auxiliary model.** For our method in the main experiments, the auxiliary biased model is simultaneously trained with the debiased model. To analyze the performance of the pretrained biased model, we equip SelecMix with the pretrained biased contrastive model and evaluate the performance of our method with the varying total number of epochs for pretraining. The solid line represents the unbiased accuracy of our method with the pretrained biased model. The dotted line represents the unbiased accuracy of our method, i.e., simultaneous training of the auxiliary biased model. As shown in Fig. 5, vanilla SC loss exhibits the high variance of the performance with respect to the number of the epochs for pretraining, compared to the proposed GSC loss.

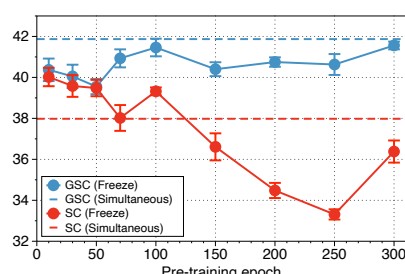

Figure 5: Comparison of simultaneous training vs. pretraining of the auxiliary model.

## 4.3 Detailed analysis of the selective mixup strategy of SelecMix

**Ablation study.** As shown in Table 4, we obtain the best performance when utilizing both the contradicting positives (A) and the contradicting negatives (B) all together in most cases. This is because it generates more diverse bias-conflicting examples compared to when using only one of them alone. On the other hand, standard mixup degrades the generalization capability in many cases which implies that the selection of the pairs is crucial.

**Comparison with a mixup variant when bias label is available.** We compare SelecMix with the recently proposed LISA [35], the mixup strategy for addressing domain generalization and subpopulation shift problems. LISA similarly applies the vanilla mixup on the pair of (i) the samples with the same label but different domains, and (ii) the samples with different labels but the same domain. Here, the domain label corresponds to the bias label in our experiment. The key differences between SelecMix and LISA are (i) ours does not require the bias label, (ii) we do not interpolate the labels of the generated samples, and (iii) we sample $\lambda$ and let $\lambda \leftarrow \min(\lambda, 1 - \lambda)$, i.e., we always assign the higher weight to the selected sample and the lower weight to the query sample (Sec. 3.3). Table 5 empirically confirms the better performance of SelecMix, using the explicit bias label for a fair comparison, over LISA, implying that augmenting the samples close to the bias-conflicting

Table 4: Detailed analysis of SelecMix. (A) denotes SelecMix with only contradicting positives. (B) denotes SelecMix with only contradicting negatives. (AB) corresponds to the proposed SelecMix.

| Dataset | Colored MNIST | | | | Corrupted CIFAR-10 | | | | BFFHQ |
|---|---|---|---|---|---|---|---|---|---|
| Ratio (%) | 0.5 | 1.0 | 2.0 | 5.0 | 0.5 | 1.0 | 2.0 | 5.0 | 0.5 |
| Vanilla | $35.71_{\pm0.83}$ | $50.51_{\pm2.17}$ | $65.40_{\pm1.63}$ | $82.12_{\pm1.52}$ | $23.26_{\pm0.29}$ | $26.10_{\pm0.72}$ | $31.04_{\pm0.44}$ | $41.98_{\pm0.12}$ | $56.20_{\pm0.35}$ |
| + mixup | $36.90_{\pm1.87}$ | $46.19_{\pm1.93}$ | $59.44_{\pm3.61}$ | $76.28_{\pm3.51}$ | $23.53_{\pm0.80}$ | $25.48_{\pm0.50}$ | $31.88_{\pm0.42}$ | $43.41_{\pm0.56}$ | $52.80_{\pm0.40}$ |
| + Ours (A) | $58.49_{\pm0.77}$ | $74.59_{\pm0.77}$ | $83.16_{\pm1.14}$ | $89.32_{\pm0.71}$ | $\underline{37.16}_{\pm0.09}$ | $\underline{41.38}_{\pm0.41}$ | $\mathbf{48.00}_{\pm0.64}$ | $\mathbf{54.40}_{\pm0.26}$ | $66.20_{\pm1.80}$ |
| + Ours (B) | $\mathbf{71.23}_{\pm1.30}$ | $\underline{81.87}_{\pm0.94}$ | $\underline{85.91}_{\pm0.51}$ | $\underline{90.53}_{\pm0.81}$ | $35.17_{\pm0.47}$ | $37.74_{\pm0.17}$ | $42.77_{\pm0.75}$ | $49.59_{\pm0.05}$ | $\underline{70.33}_{\pm0.46}$ |
| + Ours (AB) | $\underline{70.47}_{\pm1.66}$ | $\mathbf{83.55}_{\pm0.42}$ | $\mathbf{87.03}_{\pm0.58}$ | $\mathbf{91.56}_{\pm0.17}$ | $\mathbf{38.14}_{\pm0.15}$ | $\mathbf{41.87}_{\pm0.14}$ | $\underline{47.70}_{\pm1.35}$ | $\underline{54.00}_{\pm0.38}$ | $\mathbf{71.60}_{\pm1.91}$ |
| LfF | $63.86_{\pm2.81}$ | $78.64_{\pm1.51}$ | $84.95_{\pm1.71}$ | $89.42_{\pm0.65}$ | $29.36_{\pm0.18}$ | $33.50_{\pm0.52}$ | $40.65_{\pm1.23}$ | $50.95_{\pm0.40}$ | $65.60_{\pm1.40}$ |
| + mixup | $44.30_{\pm1.03}$ | $58.22_{\pm2.71}$ | $72.44_{\pm2.10}$ | $85.28_{\pm1.39}$ | $22.71_{\pm0.60}$ | $26.32_{\pm1.10}$ | $32.67_{\pm0.46}$ | $45.16_{\pm0.92}$ | $57.53_{\pm0.64}$ |
| + Ours (A) | $57.28_{\pm1.96}$ | $74.44_{\pm0.36}$ | $84.20_{\pm0.48}$ | $90.33_{\pm0.22}$ | $\underline{38.46}_{\pm0.40}$ | $\underline{42.94}_{\pm0.44}$ | $\underline{49.32}_{\pm0.28}$ | $\underline{56.11}_{\pm0.83}$ | $67.60_{\pm2.20}$ |
| + Ours (B) | $\mathbf{70.26}_{\pm1.70}$ | $\mathbf{83.14}_{\pm0.86}$ | $\underline{86.44}_{\pm0.49}$ | $\underline{91.49}_{\pm0.31}$ | $37.15_{\pm1.31}$ | $41.82_{\pm0.98}$ | $47.01_{\pm0.34}$ | $53.68_{\pm0.02}$ | $\underline{70.40}_{\pm2.23}$ |
| + Ours (AB) | $\underline{70.00}_{\pm0.52}$ | $\underline{82.80}_{\pm0.71}$ | $\mathbf{87.16}_{\pm0.62}$ | $\mathbf{91.57}_{\pm0.20}$ | $\mathbf{39.44}_{\pm0.22}$ | $\mathbf{43.68}_{\pm0.51}$ | $\mathbf{49.70}_{\pm0.54}$ | $\mathbf{57.03}_{\pm0.48}$ | $\mathbf{70.80}_{\pm2.95}$ |

Table 5: Comparison with the mixup variants when the bias label is accessible.

| Dataset | Colored MNIST | | | | Corrupted CIFAR-10 | | | | BFFHQ |
|---|---|---|---|---|---|---|---|---|---|
| Ratio (%) | 0.5 | 1.0 | 2.0 | 5.0 | 0.5 | 1.0 | 2.0 | 5.0 | 0.5 |
| Vanilla | $35.71_{\pm0.83}$ | $50.51_{\pm2.17}$ | $65.40_{\pm1.63}$ | $82.12_{\pm1.52}$ | $23.26_{\pm0.29}$ | $26.10_{\pm0.72}$ | $31.04_{\pm0.44}$ | $41.98_{\pm0.12}$ | $56.20_{\pm0.35}$ |
| + LISA (A) | $48.95_{\pm0.75}$ | $67.10_{\pm0.77}$ | $78.28_{\pm0.99}$ | $87.04_{\pm0.72}$ | $33.29_{\pm0.65}$ | $38.62_{\pm0.31}$ | $45.79_{\pm0.13}$ | $53.41_{\pm0.27}$ | $63.20_{\pm0.92}$ |
| + Ours (A) | $58.07_{\pm1.94}$ | $72.25_{\pm0.60}$ | $82.35_{\pm0.82}$ | $90.11_{\pm0.14}$ | $36.10_{\pm0.21}$ | $40.55_{\pm0.45}$ | $46.70_{\pm0.73}$ | $53.90_{\pm0.59}$ | $67.67_{\pm1.33}$ |
| + LISA (B) | $52.32_{\pm2.30}$ | $72.97_{\pm1.22}$ | $79.74_{\pm2.25}$ | $86.44_{\pm1.10}$ | $29.56_{\pm0.93}$ | $34.23_{\pm1.34}$ | $40.47_{\pm0.67}$ | $47.61_{\pm0.64}$ | $58.93_{\pm0.50}$ |
| + Ours (B) | $72.02_{\pm1.39}$ | $81.94_{\pm1.62}$ | $86.04_{\pm0.77}$ | $88.68_{\pm2.32}$ | $35.47_{\pm0.54}$ | $39.27_{\pm0.87}$ | $43.13_{\pm1.29}$ | $48.44_{\pm0.96}$ | $77.40_{\pm2.09}$ |
| + LISA (AB) | $60.85_{\pm1.72}$ | $74.42_{\pm2.27}$ | $83.20_{\pm0.52}$ | $88.73_{\pm0.39}$ | $32.71_{\pm1.09}$ | $38.18_{\pm0.90}$ | $44.15_{\pm0.39}$ | $51.57_{\pm0.45}$ | $65.20_{\pm0.53}$ |
| + Ours (AB) | $70.57_{\pm2.86}$ | $83.38_{\pm0.53}$ | $87.22_{\pm0.70}$ | $90.23_{\pm0.58}$ | $37.02_{\pm1.05}$ | $41.66_{\pm1.10}$ | $48.35_{\pm0.99}$ | $53.47_{\pm0.53}$ | $75.00_{\pm0.53}$ |

samples is crucial on the biased datasets. More discussions on the comparison between the SelecMix (using the explicit bias label) and LISA is provided in Appendix C.3.

# 5 Conclusions

We presented SelecMix, a method for debiased learning that augments bias-conflicting examples using mixup on contradicting pairs of examples. The selection of these pairs is the critical part of the method. It relies on an auxiliary model trained with a contrastive loss designed to amplify reliance on the biased features. The biased features are assumed to be "easy to learn" and incorporated earlier than others during training by SGD. SelecMix outperforms baselines on debiasing benchmarks and remains effective under the presence of label noise.

**Limitations.** The "easy-to-learn" assumption may not be correct, i.e., biased features are not always guaranteed to be learned faster than robust ones. Our method should only be used when this assumption holds, but it is unclear how to determine such cases a priori, and how frequent they are in real-world data [38]. In the worst case, our method could increase reliance on biased features and *worsen* robustness. Our experiments use semi-synthetic datasets where the assumption is made valid by construction. Thus, our results are not a sign of the broad real-world applicability of the method.

**Societal impact.** Our contributions should have a positive impact since our aim is to make machine learning systems more reliable. However, our method is only one step in this direction and the problems addressed should not be considered as solved.

## Acknowledgments and Disclosure of Funding

We would like to thank Sangdoo Yun for the useful discussions and Yeonji Song for the suggestions on the writing. We also like to thank the anonymous reviewers for their constructive comments.

This work was supported by the SNU-NAVER Hyperscale AI Center and the Institute of Information & Communications Technology Planning & Evaluation (2015-0-00310-SW.StarLab/10%, 2019-0-01371-BabyMind/10%, 2021-0-02068-AIHub/10%, 2021-0-01343-GSAI/10%, 2022-0-00951-LBA/10%, 2022-0-00953-PICA/50%) grant funded by the Korean government.

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
