# A Experimental details

## A.1 A detailed description of datasets

For the main experiments, we evaluate our method on the Colored MNIST, Corrupted CIFAR-10, and BFFHQ. The example images of each datasets are shown in Figs. 6 to 8. The Corrupted CIFAR-10 is a modified version of the CIFAR-10 dataset [17] which is constructed by applying different types of corruptions. Specifically, each class is corrupted with one specific type of corruption from the following: BRIGHTNESS, CONTRAST, DEFOCUS BLUR, ELASTIC

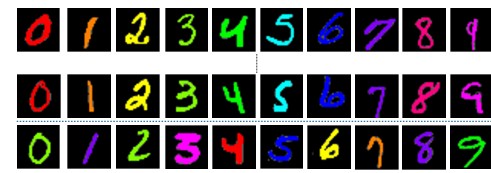

Figure 6: Example images of the Colored MNIST.

TRANSFORM, FROST, GAUSSIAN BLUR, GAUSSIAN NOISE, IMPULSE NOISE, PIXELATE, and SATURATE. For the Colored MNIST and Corrupted CIFAR-10, we report the accuracy on the unbiased test set. The BFFHQ [15] is a subset of the FFHQ dataset [12] constructed for evaluating debiasing methods. For the BFFHQ, we follow prior work [20] and report the accuracy on the test set consisting of bias-conflicting examples. The datasets are available in the official repository of DFA [20].

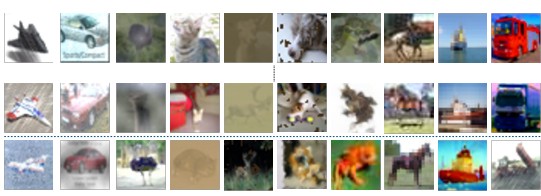

Figure 7: Example images of Corrupted CIFAR10.

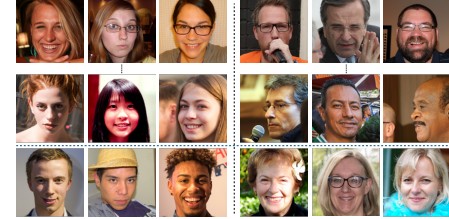

Figure 8: Example images of BFFHQ.

## A.2 Experimental details

**Setup.** We use a three-layer MLP for the Colored MNIST and a ResNet18 [9] for {Corrupted CIFAR-10, BFFHQ} as the backbone architecture for all methods. We set the batch size to 256 and 64 for {Colored MNIST, Corrupted CIFAR-10} and {BFFHQ}, respectively. We train the models for 200 and 300 epochs for {Colored MNIST, BFFHQ} and {Corrupted CIFAR-10}, respectively. To train our debiased model, we use the Adam optimizer with learning rates of 0.005, 0.001, and 0.0001 for the Colored MNIST, Corrupted CIFAR-10, and BFFHQ, respectively. To train our auxiliary contrastive model, we use the Adam optimizer with learning rates of 0.01 and 0.001 for the Colored MNIST and Corrupted CIFAR-10, respectively, and use the SGD with a learning rate of 0.4 for the BFFHQ. We did not use any additional augmentations to train the auxiliary model. Most of the experiments were run on a single RTX 3090 GPU.

**Implementation details.** In the experiments, we combine our method with the vanilla ResNet18 (V+Ours) and LfF (L+Ours). To train the debiased model $f_\theta$, we update $\theta$ with $\mathcal{L}_{total} = \lambda_{base}\mathcal{L}_{base}(\mathcal{B}) + \lambda_{ours}\mathcal{L}_{CE}(\widetilde{\mathcal{B}})$, where $\mathcal{B} = \{(x_i, y_i)\}_{i=1}^{B}$ is the batch, $\widetilde{\mathcal{B}} = \{(\tilde{x}_i, \tilde{y}_i)\}_{i=1}^{B}$ is the batch of the mixup samples (See Alg. 1), and $\mathcal{L}_{base}$ is the update rule of the base algorithm for the debiased model (e.g., CE loss for the vanilla ResNet18). We used $(\lambda_{base}, \lambda_{ours}) = (0, 1)$ for V+Ours, which corresponds to line 7 in Alg. 2 (i.e., update $\theta$ with $\mathcal{L}_{CE}(\widetilde{\mathcal{B}})$) and $(\lambda_{base}, \lambda_{ours}) = (1, 0.1)$ for L+Ours. Note that the training of the auxiliary biased model is agnostic of the base algorithm (e.g., Vanilla or LfF). We fixed the temperature $\tau = 0.2$ (hyperparameter of the contrastive loss) and $q = 0.7$ (hyperparameter of the GCE loss and GSC loss) for all experiments. For the major hyperparameters (i.e., $\lambda_{base}$, $\lambda_{ours}$, and $\tau$) of our method, we used the fixed values across the datasets and varying ratios. For the hyperparameters of the baselines, we used their default choice, even if the optimal values may vary for the different ratios in the same dataset. Since there is no provision for an unbiased validation set

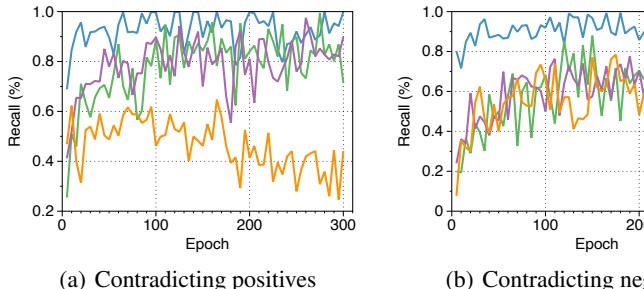

(a) Contradicting positives       (b) Contradicting negatives

Figure 9: Accuracy of discovering contradicting pairs. High recall implies that most of the contradicting pairs in the batch are discovered by SelecMix. The cos and $l_2$ denote respectively cosine and $l_2$ distance on the embedding space learned with the GCE loss. KLD denotes the KL-divergence of the softmax outputs of the prediction head trained with the GCE loss.

in most existing benchmarks, we follow the evaluation protocol of prior works [1, 20, 24] and report the best accuracy (i.e., an "oracle" model selection). Experimental results on the Corrupted CIFAR10 and the BFFHQ have averaged over 3 independent trials. For the Colored MNIST, we report the average test accuracy over 5 independent trials.

**Label noise experiment.** For the label noise experiments in Fig. 3, we modify the Colored MNIST ($\alpha = 1\%$) and the Corrupted CIFAR-10 ($\alpha = 1\%$) by replacing the label with a random one (i.e., label noise) for a portion $\beta \in \{0.5\%, 1\%, 2\%, 5\%\}$ of the training examples. We used the same hyperparameters for the label noise experiments and the main experiments.

# B  Additional experiments

## B.1  Identifying contradicting pairs

We compare our biased model with the model trained with the GCE loss, in terms of the accuracy of identifying contradicting positives and negatives on the Corrupted CIFAR-10 dataset. As shown in Fig. 9, our biased contrastive model better identifies the contradicting pairs, especially for discovering contradicting negatives. This implies that the GSC loss effectively clusters the examples with respect to the biased features, which corroborates the results of Table 2.

## B.2  Bias-conflict accuracy

In this subsection, we provide the bias-conflict accuracy (i.e., the accuracy of the bias-conflicting examples) of the baselines and our method. As shown in Table 6, our method outperforms the baselines in terms of the bias-conflict accuracy as well.

Table 6: Bias-conflict accuracy of the baselines and our method.

| Dataset | Vanilla | Mixup [37] | EnD [31] | LfF [24] | DFA [20] | V+Ours | L+Ours |
|---|---|---|---|---|---|---|---|
| Colored MNIST (1.0%) | 44.14±2.71 | 38.80±3.85 | 62.92±2.56 | 71.40±2.63 | 76.06±3.15 | **77.38**±1.87 | 74.62±0.94 |
| Colored MNIST (2.0%) | 59.32±1.52 | 51.88±3.56 | 76.31±1.16 | 80.82±2.75 | 81.39±1.97 | **83.32**±1.57 | 82.70±0.73 |
| Corrupted CIFAR10 (1.0%) | 16.30±0.50 | 14.77±0.50 | 16.22±0.53 | 24.08±0.97 | 25.00±2.55 | 35.89±1.05 | **36.17**±1.25 |
| Corrupted CIFAR10 (2.0%) | 21.40±0.74 | 22.23±0.89 | 23.77±0.22 | 34.00±1.06 | 31.84±1.09 | **44.24**±1.07 | 43.88±1.41 |

## B.3  Comparison with JTT

In this subsection, we provide a comparison with JTT [22]. Table 7 shows the test accuracy of JTT and of our method. We used grid-search to choose the hyperparameters of JTT, i.e., the first stage training epoch $T$ and the upsampling ratio $\lambda$ as follows: $T \in \{1, 5, 10, 20, 30, 50\}$ and $\lambda \in \{5, 10, 30, 50, 100\}$.

Table 7: Comparison with JTT [22].

| Dataset | Vanilla | JTT [22] | V+Ours |
|---|---|---|---|
| Colored MNIST (1.0%) | $50.51_{\pm 2.17}$ | $62.35_{\pm 3.30}$ | $\mathbf{83.55}_{\pm 0.42}$ |
| Colored MNIST (2.0%) | $65.40_{\pm 1.63}$ | $74.04_{\pm 1.33}$ | $\mathbf{87.03}_{\pm 0.58}$ |
| Corrupted CIFAR10 (1.0%) | $26.10_{\pm 0.72}$ | $28.55_{\pm 0.27}$ | $\mathbf{41.87}_{\pm 0.14}$ |
| Corrupted CIFAR10 (2.0%) | $31.04_{\pm 0.44}$ | $33.03_{\pm 0.52}$ | $\mathbf{47.70}_{\pm 1.35}$ |
| BFFHQ | $56.20_{\pm 0.35}$ | $58.40_{\pm 0.35}$ | $\mathbf{71.60}_{\pm 1.91}$ |

### B.4 Experiments on the UTKFace dataset

**Dataset.** The UTKFace dataset is composed of various human face images. The dataset contains labels for race, age, and gender. Fig. 10 shows the example images of the UTKFace dataset. Following the prior work [11], we divide the samples into two groups for each label and compose a biased sub-dataset. We defer the details to [11].

**Training details.** For the debiased model, we use a ResNet18 pretrained on the ImageNet dataset as a backbone architecture and use the Adam optimizer with a weight decay of 0.0001. We set an initial learning rate of 0.001 when the label is either gender or race, and 0.0005 when the label is age. We use the mixup of the contradicting positives for SelecMix. We

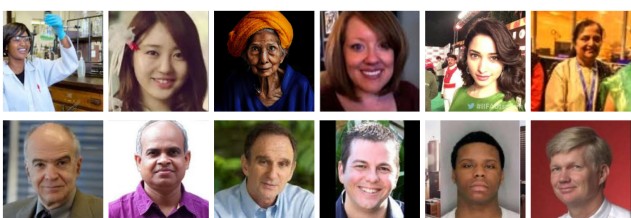

Figure 10: Example images of the UTKFace dataset.

use $(\lambda_{\text{base}}, \lambda_{\text{ours}}) = (1, 0.1)$. The learning rate is decayed by a factor of 10 at 1/3 and 2/3 of the total training epochs, following the prior work [11]. For the auxiliary contrastive model, we use ResNet18 with the SGD optimizer. We train the models for 40 epochs with a batch size of 128. We use the data augmentations of random resized crop and random horizontal flip.

**Experimental results.** As shown in Table 8, our method consistently outperforms the baselines for various compositions of target and bias features. This implies that the proposed method successfully scales to the complex real-world datasets which the prior works struggles on.

Table 8: Unbiased accuracy of the baselines and our method trained on UTKFace dataset.

| (Target, Bias) | Vanilla | LfF [24] | DFA [20] | V+Ours |
|---|---|---|---|---|
| (Gender, Age) | $\underline{74.09}_{\pm 0.99}$ | $67.93_{\pm 2.48}$ | $70.34_{\pm 0.85}$ | $\mathbf{77.00}_{\pm 0.66}$ |
| (Age, Gender) | $\underline{92.00}_{\pm 0.35}$ | $84.90_{\pm 1.72}$ | $84.23_{\pm 1.66}$ | $\mathbf{92.07}_{\pm 0.10}$ |
| (Race, Gender) | $\underline{77.73}_{\pm 0.23}$ | $70.72_{\pm 2.46}$ | $70.56_{\pm 0.89}$ | $\mathbf{78.58}_{\pm 0.49}$ |
| (Gender, Race) | $\underline{88.68}_{\pm 0.21}$ | $77.79_{\pm 1.43}$ | $82.03_{\pm 2.36}$ | $\mathbf{88.92}_{\pm 0.11}$ |

## C Additional discussions

### C.1 Extended related work

JTT [22] proposed a 2-stage training framework to improve the group robustness without the group information. In the first stage, they train the vanilla network with ERM and construct the error set consists of the misclassified samples. In the second stage, they oversample examples of the error set to train the second network. JM1 [8] also constructs the error set and utilizes class-conditional mixup to further improve the group robustness. In the transfer learning setting, Bao et al. [3] learns spurious features from multiple environments and constructs the metric space based on the spurious features. In contrast, our method exploits the easy-to-learn heuristic and learns the hypersphere embedding space with the proposed GSC loss.

We now discuss the difference between JM1 [8] and our method, in particular, SelecMix (A). While class-conditional mixup is utilized in both our method and JM1, the key difference is in the sampling

of the pairs for the mixup. They combine the misclassified sample from the vanilla model with the sample that has the same label randomly chosen from the rest. On the other hand, SelecMix (A) combines the contradicting pair having the same label but *the most dissimilar* biased features, by measuring the similarity with the biased model.

## C.2   Discussion on the relationship between GCE and GSC losses

In this subsection, we discuss the relationship between the generalized cross-entropy (GCE) loss [40] and the proposed GSC loss. To begin with, we first explain how the GCE loss amplifies the reliance of the auxiliary model on biased features, compared to the standard cross-entropy (CE) loss.

The GCE loss is defined as $\mathcal{L}_{\text{GCE}}(\boldsymbol{p}, y) = (1 - \boldsymbol{p}_y^q) / q$ where $\boldsymbol{p}$ is the softmaxed vector of predictions from the model and $\boldsymbol{p}_y$ its $y^{\text{th}}$ component, $y$ is the ground truth class ID, and $q \in (0, 1]$ a scalar hyperparameter.

The GCE loss simplifies to the CE loss as $q \to 0$. Assuming that the predictions are produced by a model of parameters $\boldsymbol{\theta}$, the gradients of the GCE and CE losses are related as follows:

$$\frac{\partial}{\partial \theta} \mathcal{L}_{\text{GCE}_\theta}(\boldsymbol{p}, y) \;=\; \boldsymbol{p}_y^q \cdot \frac{\partial}{\partial \theta} \mathcal{L}_{\text{CE}_\theta}(\boldsymbol{p}, y). \tag{6}$$

Here, the term $\boldsymbol{p}_y^q$ assigns the higher weight to the samples with a high probability $\boldsymbol{p}_y$, thus upweights the "easy" samples and amplifies the reliance on the biased features. While the model trained with the CE loss also focuses on the biased features since they are learned first, the GCE loss was shown to be more effective in identifying bias-conflicting examples by Nam et al. [24]. Similar to the GCE/CE, our GSC loss improves over the SC loss [13] in encouraging the model to rely primarily on biased features. The term $\hat{p}_{i,k}^q$ in Eq. (3) plays the same role as the term $\boldsymbol{p}_y^q$ in the GCE.

## C.3   A discussion comparing SelecMix with explicit bias labels and LISA

As described in Sec. 4.3, LISA [35] is a similar mixup strategy that applies mixup on contradicting positives and negatives. When the explicit bias label is available, SelecMix also directly picks the contradicting pairs and applies mixup. Here, the key difference is that (i) SelecMix does not mix the labels, and (ii) SelecMix always assigns the higher weight to the selected sample. Thus, SelecMix more explicitly augment various bias-conflicting samples, compared to LISA. As shown in Table 5, the performance gap between SelecMix and LISA is significant for the small $\alpha$. This also implies that SelecMix is effective for debiased learning on the highly biased dataset.