# OpenReview forum: "SelecMix: Debiased Learning by Contradicting-pair Sampling"
_NeurIPS.cc/2022/Conference — NeurIPS 2022 Accept_

### Official Review · Reviewer_HQZ4 · 2022-07-03

**Rating:** 6
**Confidence:** 5
**Soundness:** 2 fair
**Presentation:** 2 fair
**Contribution:** 3 good

**Summary:**

The authors proposed selective mixup for learning robust classifiers. Specifically, they apply mixup to
1. input pairs that have the same label but different bias attribute values.
2. input pairs with different label but with similar bias attribute values.
Since the bias attribute values are usually not given, they estimate it by a bias-amplified model (that favors easy-to-learn shallow features during training). On MNIST, CIFAR10 and BFFHQ, combining selective mixup with previous methods consistently improves the robust performance.

**Questions:**

1. I think comparing with JTT [1] is necessary and will strengthen the paper.

**Ethics Review Area:**

["I don’t know"]

**Limitations:**

Yes.

**Strengths And Weaknesses:**

Strength:
1. Adapting input mixup for de-biasing is interesting.
2. Consistently good performance across three datasets.

Weakness:
1. The idea is clearly presented (nice figure) but the paper is poorly written.
    + For example, you cannot just simply name *bias attributes* as *bias*. Bias often refers to unwanted correlations between the input and the output. The sentence "the same label but dissimilar bias [line 11]" is just very confusing.
2. A few missing references.
    + [1] uses group DRO to learn a robust classifier from the training mistakes. It outperforms LfF by a significant margin, I am very curious to see how does SelecMix compare to this method.
    + [2] proposes to use a biased model to guide the training of a debased model. I think they are one of the first among this line and it should definitely be cited.
    + [3] uses different data environments to learn a metric space that encodes examples based on their bias attributes.


[1] Liu, Evan Z., et al. "Just train twice: Improving group robustness without training group information." International Conference on Machine Learning. PMLR, 2021.

[2] Sanh, Victor, et al. "Learning from others' mistakes: Avoiding dataset biases without modeling them." International Conference on Learning Representations. 2020.

[3] Bao, Yujia, Shiyu Chang, and Regina Barzilay. "Learning Stable Classifiers by Transferring Unstable Features." ICML 2022.

---

> ### Author Response · Authors · 2022-08-02
> **Response to Reviewer HQZ4**
>
> We sincerely appreciate your efforts and constructive comments to improve the manuscript. We respond to each of your comments below:
>
> ---
>
> > I think comparing with JTT [1] is necessary and will strengthen the paper.
> >
> - We conducted the experiment for JTT [1] and we provide the comparison with our method. As shown in the table below, our method outperforms JTT on debiasing benchmarks. Note that we used grid-search to choose the hyperparameters of JTT: $T\in\\{1, 5, 10, 20, 30, 50\\}$ and $\lambda\in\\{5, 10, 30, 50, 100\\}$. We will incorporate this and further results in the final draft.
>
> $$
> \begin{array}{cccc}
> \hline
> \text{Dataset}
> & \text{Vanilla}
> & \text{JTT}
> & \text{V+Ours}
> \\\
> \hline
> \hline
> \text{Colored MNIST } (1.0 \\% ) & {50.51}\pm{2.17} & {62.35}\pm{3.30} & \textbf{83.55}\pm{0.42} \\\
> \text{Colored MNIST } (2.0 \\% ) & {65.40}\pm{1.63} & {74.04}\pm{1.33} & \textbf{87.03}\pm{0.58} \\\
> \hline
> \text{Corrupted CIFAR10 } (1.0 \\% ) & {26.10}\pm{0.72} & {28.55}\pm{0.27} & \textbf{41.87}\pm{0.14} \\\
> \text{Corrupted CIFAR10 } (2.0 \\% ) & {31.04}\pm{0.44} & {33.03}\pm{0.52} & \textbf{47.70}\pm{1.35} \\\
> \hline
> \text{BFFHQ } & {56.20}\pm{0.35} & {58.40}\pm{0.35} & \textbf{71.60}\pm{1.91} \\\
> \hline
> \end{array}
> $$
>
> ---
>
> > Suggestions in writing
> >
> - Thank you for the helpful suggestions. We will incorporate your suggestions and revise the manuscript to improve the clarity, including (1) changing the simplified term *bias* and (2) additional references.
>
> ---
>
> [1] Liu, Evan Z., et al. "Just train twice: Improving group robustness without training group information." *International Conference on Machine Learning*. PMLR, 2021.

---

> > ### Comment · Reviewer_HQZ4 · 2022-08-07
> > **reply to the rebuttal**
> >
> > I appreciate the authors for addressing my questions. The added comparison against (JTT) is very strong. I also have read other reviewers' comments. Accordingly, I am raising my score from 5 to 6.

---

### Official Review · Reviewer_YfvZ · 2022-07-07

**Rating:** 4
**Confidence:** 1
**Soundness:** 2 fair
**Presentation:** 2 fair
**Contribution:** 2 fair

**Summary:**

This paper proposes SelectMix for debiased laerning. Contradicting pairs are selected for mixup with different training loss to achieve the learning goal. The experiments show improvement over previous methods.

**Questions:**

see above

**Limitations:**

Limitations and societal impact are presented.

**Strengths And Weaknesses:**

Strengths
1. Comprehensive experiments and analysis are provided. The results also show improvement.

Weaknesses
1. The organization of the paper is not very easy to follow.
2. The core contribution of the paper is SelectMix, but also a contrastive objective is included in training. The analysis and improvement of adding the contrastive loss is not shown.

---

> ### Author Response · Authors · 2022-08-02
> **Response to Reviewer YfvZ**
>
> Thank you for your invaluable review. We initially address your concern below:
>
> ---
>
> > The analysis and improvement of adding the contrastive loss is not shown.
> >
>
> We provided the detailed analyses of the proposed contrastive loss (i.e., $\mathcal{L}_\textit{G-SupCon}$) in **Section 4.2**. The key point is that $\mathcal{L}_\textit{G-SupCon}$ **amplifies the reliance on bias features** of the model so that **the feature clustering in the embedding space serves as a good indicator of the similarity of bias features**. To be specific,
>
> - **Table 2** shows that SelecMix achieves the best performance when the similarity is measured with the model trained with $\mathcal{L}_\textit{G-SupCon}$ (i.e., Ours), compared to GCE loss (i.e., cos, $l_2$, and KL-divergence). The visualization of the feature clustering is shown in **Figure 4**.
> - **Table 3** shows that $\mathcal{L}_\textit{G-SupCon}$ outperforms GCE loss on predicting the bias labels, which also illustrates the effectiveness of the proposed loss on amplifying the reliance on the bias features.
>
> ---
>
> Please let us know if you have any remaining concerns.
>
> Sincerely, Authors

---

### Official Review · Reviewer_b6K4 · 2022-07-09

**Rating:** 7
**Confidence:** 4
**Soundness:** 3 good
**Presentation:** 3 good
**Contribution:** 3 good

**Summary:**

This work deals with the problem of groups (bias) robustness under distribution shift between train and test set. The authors propose a mixup strategy where samples with i) same label but different groups, and samples with ii) different labels but similar bias are mixed without having access to bias (or group) annotation. Cosine similarity is used as a proxy to find clusters of different groups training an additional model via supervised contrastive learning. The authors provide experiments on vision datasets to corroborate their method.

**Questions:**

* Mixup-based augmentation has been employed to handle distribution shifts finetuning large models, for example in this recent work - https://openreview.net/forum?id=HI2ilxFli0W - class-conditional mixup is employed to improve group robustness in presence of group shift between train and test in absence of group annotation. How would your method, in particular, `i) the same label but dissimilar bias` part of SelecMix, compare to this or a similar approach for transfer learning or distribution shift?

* For SelecMix, other than i) you also mix `ii) samples with different labels but similar bias`. How important is this second mixing for SelecMix to work? I would assume that using standard mixup on a train set with highly unbalanced groups (99% vs 1% or 95% vs 5%) the results should be identical to applying ii) and shouldn't help group robustness. Can you elaborate on why using ii) is helpful compared to just applying mixup in Table 4?

* Do you think you could use your method to de-bias language or a different modality?

* What is the metric in Table 1? Average accuracy? It would be interesting to see the worst-group (label, bias) accuracy for the different methods.

**Limitations:**

* The effectiveness of the method is based on how well the cosine similarity can cluster samples from the same group together. This is a reasonable assumption for images but not necessarily for other modalities, and for this reason, the method is (or could be) limited in applicability.
* Additionally I think this approach will have issues with larger datasets with more variety and number of classes and biases.

**Strengths And Weaknesses:**

strengths

* Important work that explores the use of mixup strategy in distribution shift with possible applications to foundational models.
* The paper is well written and clear with a solid set of experiments and ablation studies.
* The method is effective and experiments are provided on vision benchmarks.

weaknesses

* Cosine similarity to cluster samples from the same group can be ineffective for more complex datasets with a large number of classes or a large variety of biases.
* Method limited to images.

---

> ### Author Response · Authors · 2022-08-02
> **Response to reviewer b6K4**
>
> We sincerely appreciate your efforts and constructive comments to improve the manuscript. We respond to each of your comments below. For brevity, we will denote (i) mixing contradicting positive pairs as **SelecMix-A** and (ii) mixing contradicting negative pairs as **SelecMix-B**.
>
> ---
>
> > How important is this second mixing for SelecMix to work? […] Can you elaborate on why using ii) is helpful compared to just applying mixup in Table 4?
> >
> - The standard mixup applied on the highly biased dataset is likely to combine bias-aligned samples. Thus, generated mixup sample does not contradict the prediction based on the bias attribute.
> - In contrast, for each instance $(x_i, y_i)$ in the current mini-batch (which is likely to be bias-aligned), SelecMix-B identifies another instance $(x_k, y_k)$ which have different label but similar bias features (thus likely to be bias-conflicting) for mixup pair. Here, another key difference between SelecMix-B and the standard mixup is that **we do not mixup the labels**, i.e., $(\widetilde{x}_i, \widetilde{y}_i) \leftarrow (\lambda x_i + (1-\lambda) x_k, y_k)$, and assign higher weight to $x_k$. This can be viewed as **injecting bias-aligned samples as noise to existing bias-conflicting samples**. Since $x_i$ and $x_k$ share similar bias feature, $\widetilde{x}_i$ is also assumed to do so. Thus, the generated sample $(\widetilde{x}_i, \widetilde{y}_i)$ acts as bias-conflicting sample since it has the same label and similar bias features to the existing bias-conflicting sample $(x_k, y_k)$.
> - By utilizing both SelecMix-A and SelecMix-B altogether, we can generate more diverse bias-conflicting samples compared to when applying SelecMix-A or SelecMix-B alone. Table 4 shows an improved performance when applying both of them together in most cases.
>
> ---
> > How would your method, in particular, i) the same label but dissimilar bias part of SelecMix, compare to JM1 [1]?
> >
> - Class-conditional mixup is utilized in both our method and JM1 [1]. However, the key difference between our method and JM1 is on the sampling of the pairs for mixup. JM1 [1] combines the misclassified sample from the vanilla model with the sample that have the same label randomly chosen from the rest. On the other hand, SelecMix-A combines the contradicting pair having the same label but the **most dissimilar** bias features, by measuring the similarity with the biased model. We will add this discussions in the revised manuscript.
>
> ---
> > What is the metric in Table 1? It would be interesting to see the worst-group (label, bias) accuracy for the different methods.
> >
> - Table 1 shows the accuracy on the unbiased test sets.
> - We observe that the worst-group accuracy among all 100 groups (label, bias) in Corrupted CIFAR-10 is mostly 0% for many baselines. As an alternative, we report the *bias-conflict accuracy* (i.e., the accuracy of minor group samples) which is often used as a sub-metric for the analysis [2]. As shown in the table below, our method outperforms baselines in terms of bias-conflict accuracy as well. We will incorporate this and further results in the final draft.
>
> $$
> \begin{array}{cccccccc}
> \hline
> \text{Dataset}
> & \text{Vanilla}
> & \text{Mixup}
> & \text{EnD}
> & \text{LfF}
> & \text{DFA}
> & \text{V+Ours}
> & \text{L+Ours}
> \\\
> \hline
> \hline
> \text{Colored MNIST } (1.0 \\% ) & {44.14}\pm{2.71} & {38.80}\pm{3.85} & {62.92}\pm{2.56} & {71.40}\pm{2.63} & \underline{76.06}\pm{3.15} & \textbf{77.38}\pm{1.87} & {74.62}\pm{0.94} \\\
> \text{Colored MNIST } (2.0 \\% ) & {59.32}\pm{1.52} & {51.88}\pm{3.56} & {76.31}\pm{1.16} & {80.82}\pm{2.75} & {81.39}\pm{1.97} & \textbf{83.32}\pm{1.57} & \underline{82.70}\pm{0.73} \\\
> \hline
> \text{Corrupted CIFAR10 } (1.0 \\% ) & {16.30}\pm{0.50} & {14.77}\pm{0.50} & {16.22}\pm{0.53} & {24.08}\pm{0.97} & {25.00}\pm{2.55} & \underline{35.89}\pm{1.05} & \textbf{36.17}\pm{1.25} \\\
> \text{Corrupted CIFAR10 } (2.0 \\% ) & {21.40}\pm{0.74} & {22.23}\pm{0.89} & {23.77}\pm{0.22} & {34.00}\pm{1.06} & {31.84}\pm{1.09} & \textbf{44.24}\pm{1.07} & \underline{43.88}\pm{1.41} \\\
> \hline
> \end{array}
> $$
>
> ---
> > Do you think you could use your method to de-bias language or a different modality?
> >
> - We speculate that our method can be applied to different modalities as long as mixup is applicable. We consider this as a promising future direction.
>
> ---
> [1] Giannone et al. "Just Mix Once: Mixing Samples with Implicit Group Distribution." *NeurIPS 2021 Workshop on Distribution Shifts: Connecting Methods and Applications*. 2021.
>
> [2] Hong et al. "Unbiased classification through bias-contrastive and bias-balanced learning." In NeurIPS 2021.

---

> > ### Comment · Reviewer_b6K4 · 2022-08-05
> > **Comments**
> >
> > Thanks for the rebuttal. I appreciate the explanations and detailed answers to my questions. I understand better the paper and problem framing. I am raising my score and score for contribution.

---

### Author Response · Authors · 2022-08-09
**Revision Summary**

We really appreciate the reviewers’ time and effort. We followed all recommendations and included the suggested discussions, as well as additional experiments in Appendix A.8-A.10. We also improved clarity and rigor throughout the writing. Updates are highlighted in blue.

---

### Meta-Review · Area_Chair_mUrC · 2022-08-29

**Recommendation:** Accept
**Confidence:** Less certain

**Metareview:**

This paper proposes to handle so-called spurious / undesirable signals that are correlated with but do not entail the label (and where it is considered undesirable that a classifier should rely on these signals). The authors propose a variation of the mixup training heuristic where for each example, one selects a bias-conflicting pair. Because the bias-conflicting pairs are rare, they are oversampled to form the mixup pairs and intuitively, this makes the biased signal less predictive. The authors compare against other methods in the case where the "spurious feature" is known and propose a further heuristic for automatically providing pseudo "bias labels" based on the intuition that spurious features of concern are often "easy to learn" and thus examples tend to be grouped together by their spurious features earlier in training. This seems to work well on some toy datasets but the degree to which guesses are piled upon guesses here is of concern. Overall, this is a borderline paper, with 2 of 3 reviewers liking it and one championing it for acceptance.

**Award:**

No

---

### Decision · Program_Chairs · 2022-09-14

Accept